# A LEARNING-AUGMENTED OVERLAY NETWORK

## ABSTRACT

This paper studies the integration of machine-learned advice in overlay networks to improve the overall connectivity. Our algorithms are based on Skip List Networks (SLN), which is natural extension of skip lists that supports pairwise communication. In particular, our work goes beyond learning-augmented single-source skip lists (studied recently in ICLR 2025 by Fu et al. (2025) and ICML 2024 by Zeynali et al. (2024)), considering a prediction model where each node of the network individually receives a local prediction of its future communications to the rest of the network. We utilize this model to develop a distributed, learning-augmented SLN to optimize the serving of any weighted pairwise demand.

We first solve the optimization problem of finding an optimal SLN given a certain demand, which we show is polynomial with a dynamic programming approach. We then introduce a novel network structure called Continuous SLN, where the heights of each node is relaxed to be any real number. Finally, we show how a random, uniform noise on top of each node's height makes the network robust against any predictions, even adversarial, while the performances are kept unchanged when the predictions are desired. Concretely, adversarial predictions can cause our network to be a logarithmic factor away from any optimal network without prediction. Furthermore, we show that, for highly sparse demands, a refined version of our algorithm shows no drawbacks in asymptotics for any prediction and presents exponential improvements when the predictions are good. Finally, we empirically show that our learning-augmented overlay network demonstrate resistance against small error with evaluations on synthetic and real-world data-sets.

## 1 INTRODUCTION

Overlay networks are one of the key technological advancements that allowed decentralized and distributed systems to scale. A vast body of work has been devoted to optimize the efficiency of these networks Lua et al. (2005); Ratnasamy et al. (2001); Aspnes & Shah (2003); Harvey et al. (2003); Kiffer et al. (2021). Traditionally, the efficiency of an overlay network is estimated with two key metrics which are the *routing path length* between a pair of nodes and the *maximum degree* of each node in the network. State-of-the-art overlay networks achieve both a logarithmic routing path length and a logarithmic maximum degree Stoica et al. (2003); Fraigniaud & Gauron (2006).

Up until recently, most work on overlay networks focused on demand-oblivious *worst-case* scenarios, where no prior information about the possible future demand is known. However, a new line of work studies *demand-aware* networks Schmid et al. (2016); Avin et al. (2020b); Figiel et al. (2025) which can adapt their connectivity properties to better fit the current traffic. Noting that most networking traffics are *sparse* and partly *predictable* Avin et al. (2020a), the idea is to detect the few highly active nodes and move them topologically closer to reduce the overall delays. The model includes a *demand matrix* which indicates the amount of traffic between any two nodes in the network; A goal of a demand-aware protocol is then to minimize the *total weighted path length* Schmid et al. (2016).

This paper presents a peer-to-peer (P2P) overlay network design where each node can adapt its set of neighbors to better fit the future traffic demands. We do not assume that the demand matrix is known by the nodes; Rather, we stick with the distributed nature of overlay networks and assume that each node has access to a local *prediction* about the future demands — the prediction may only consist of partial information about the future traffic, an integer number in the case of this work. In our model, a node may obtain a new prediction at any time to use at its own discretion, independently from the other nodes. Crucially, the predictions are *untrusted* and we evaluate our overlay network in the

*learning-augmented* setting with two metrics: *consistency* and *robustness* Lykouris & Vassilvitskii (2018). The consistency of the algorithm is its performance when the predictions are always perfect, its robustness is its performance when the predictions are worst possible.

## 1.1 OUR SETUP

**Network and Demand.** We model a network with an undirected graph with $n$ nodes. An edge of the graph means that the two end-nodes can communicate directly without the help of other nodes. The graph is an abstract representation of the routing possibilities between the nodes and may be different from the underlying, physical network that carries the communications.
At the considered time, we model a *demand* between the nodes with with a *demand matrix* $W \in M_n(\mathbb{R})$, where $W(u, v)$ is the frequency of communications from $u$ to $v$.

**P2P Network Protocols.** A peer-to-peer (P2P) network protocol is an algorithm distributed over multiple nodes that provides routing facilities. On top of the physical, static network, the P2P protocol keeps track of local routing information stored inside each node.

**Skip Lists and SLN.** A skip list Pugh (1990) is a well-known data structure that consists of a set of sparser and sparser chained lists. Each node $u$ has an integer *height* parameter $h_u$ in $[1, \lceil \log n \rceil]$ that equals the number of lists the node belongs to. Skip lists come with a simple node-retrieval algorithm that starts from the pointer of the data structure (node zero) and gradually dive into the denser lists to find the queried item. The query time is provably optimal (for data structures with average constant degree) when the heights are well-chosen, and optimal with high probability when the heights are drawn from a geometric distribution.

A Skip List Network (SLN) Mandal et al. (2015) is a P2P design based skip lists which establishes the routing informations solely based on the heights and ids of the nodes. This paper studies P2P network protocols in which each node $u$ holds a *height* variable $h_u$ and a unique identifier which defines a total order at any time: for any two different nodes $u$ and $v$, either $u < v$ or $v < u$. In a SLN, two nodes $u$ and $v$ form an edge, i.e., store each other's id, if and only if there is no node $x$ such that $u < x < v$ and $h_x \geq \min(h_u, h_v)$ (Definition 1).

**Prediction Model.** Our prediction model is the same as in Aradhya & Scheideler (2025). The origin of the predictions is left intentionally unclear; it can be a central server that aggregates telemetry data, predicts the demand and distributes the optimal heights to each node, or it can be a value computed by each node itself based on the observed traffic. Most importantly, we assume that the predictions are (1) distributed, each node receives its own and uses it at its own discretion, (2) repeated, the node should be able to modify its topology at each update of the prediction, and (3) untrusted, the predictions could be overall optimal, adversarial, and anything in between. For the P2P application, the prediction is an argument of the initialization routine that the node runs before joining the network — in case of a new prediction, the node leaves and reinitializes.

## 1.2 CONTRIBUTIONS AND TECHNICAL NOVELTY

The contributions of this paper are threefold. First, we formulate the Optimum Static Skip List Network (OSSLN) problem and prove it can be solved in polynomial time, with an algorithm based on dynamic programming (Theorem 1). This result extends the previous work of Martinez & Roura (1995) to the case with pairwise demands in line with recent demand-aware results Schmid et al. (2016). This algorithm demonstrates that the task of finding optimal predictions for our learning-augmented P2P protocol is polynomial, which allowed us to run our evaluations.

As our second contribution, we present a new P2P design called *Continuous Skip List Network* (Continuous SLN, Definition 8), which generalizes Skip List Networks Avin et al. (2020c) (SLN, Definition 1) and skip lists Pugh (1990). We show that a SLN with real-valued heights enables a functioning P2P protocol in that we implement the three basic primitives route, join and leave with state-of-the-art, logarithmic message complexity. We note that all known P2P designs that are based on skip lists Harvey et al. (2003); Fu et al. (2025); Aspnes & Shah (2003) historically perceive it as chained lists stacked on each other and explicitly use that discrete nature in the protocol implementation, we show that this is not necessary. We then present a Continuous SLN protocol

called UNIFORM that draws its heights uniformly at random in the interval $(0, 1)$ and show that UNIFORM matches the performances of state-of-the-art P2P protocols with both routing path lengths and maximum degrees lower than $O(\log n)$ with high probability.

As our third contribution, we present a learning-augmented P2P protocol, LASLIN (Learning-Augmented Skip List Network), which combines our UNIFORM protocol with a predicted SLN to strike consistency and robustness guarantees (Definition 7). For each node, LASLIN takes as prediction an integer between 1 and $\lceil \ln n \rceil$ and sets it as its *integer heights* at initialization time; it then draws a number in $(0, 1)$ uniformly at random and adds it to the integer heights to obtain its final heights as a Continuous SLN. We show that LASLIN is comparable to an optimum static SLN when the predictions are correct ($O(1)$-consistent) and at worst a $O(\log n)$ factor off a demand-oblivious P2P protocol with arbitrary predictions ($O(\log^2 n)$-robust). Finally we show that, in the special case of a very sparse demand matrix (Definition 9), the version of LASLIN with binary predictions outperforms any demand-oblivious P2P protocol regardless of the prediction quality: LASLIN obtains an exponentially better cost with correct predictions while it asymptotically matches the cost of the state-of-the-art P2P protocols for arbitrary predictions (see Theorem 9). We provide simulations to assess the performance of LASLIN on Zipfian demand matrices.

## 2    MODEL AND METRICS

In this section, we formally define our model and the metrics we use to evaluate our P2P protocols.

**Definition 1** (SLN). *A Skip List Network (*SLN*) $\mathcal{N}$ is a triple $(V, h, E)$ where:*

- *$V$ is a totally ordered set of node ids;*

- *$h$ is a height function $h : V \longrightarrow \mathbb{N}^+$;*

- *$E = \{\{u, v\} \mid \forall x \in V \ s.t. \ u < x < v, \ h_x < \min(h_u, h_v)\}$.*

**Definition 2** (Neighbor and Degree). *We say that two nodes $u$ and $v$ are* neighbors *in a* SLN *$\mathcal{N} = (V, h, E)$ if $\{u, v\} \in E$. The degree of a node $u$ is its number of neighbors.*

We redefine the greedy, local routing path proper to SLN.

**Definition 3** (Routing Path). *Let $\mathcal{N} = (V, h, E)$ be a SLN and $u, v \in V$ be two different nodes. The* routing path *from $u$ to $v$ in $\mathcal{N}$ is the sequence of edges of $E$ $s = e_1, e_2 \ldots e_k$ such that: (1) $u \in e_1$, (2) $v \in e_k$, and (3) $\forall i \in [k]$, if $u < v$ then $e_i = \{x, y\}$ where $y$ is the largest node neighbor of $x$ such that $x < y \leq v$, otherwise if $v < u$ then $e_i = \{x, y\}$ where $y$ is the smallest node neighbor of $x$ such that $v \leq y < x$.*

More intuitively, the routing path goes to the next neighbor that is the closest to the destination (in terms of ids) without "flying over" the destination.

**Definition 4** (Cost). *Let $\mathcal{N} = (V, h, E)$ be an SLN and $W$ be a demand matrix. The cost function is:*

$$Cost(\mathcal{N}, W) = \sum_{u,v \in V} W(u, v) \cdot d_{\mathcal{N}}(u, v),$$

*where $d_{\mathcal{N}}(u, v)$ is the routing path length between $u$ and $v$ in $\mathcal{N}$.*

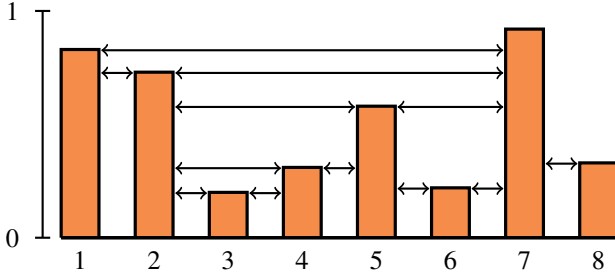

**Figure 1:** A continuous SLN with real-valued heights in $(0, 1)$.

**Definition 5** (Optimum Static Skip List Network (OSSLN) problem). *Given a set of node ids $V$ and a demand matrix $W$, the objective of the Optimum Static Skip List Network (OSSLN) problem is to find a height assignment $h : V \longrightarrow \mathbb{N}^+$ that minimizes the cost of the SLN $\mathcal{N} = (V, h, E)$.*

**Definition 6** (Learning-augmented P2P protocol). *A learning-augmented P2P protocol ALG is a P2P protocol whose initialization routine takes an additional parameter p, the prediction.*

In our model, a node that wishes to update its prediction as it already joined the network simply leaves, re-intializes and joins again.

**Definition 7** (Consistency and Robustness). *We say that a learning-augmented* SLN *protocol is*

- *$\alpha$-consistent if, for all demand matrices, there exists an assignment of predictions to the nodes such that the expected cost of the protocol is at most $\alpha$ times the cost of an optimal optimum static SLN.*

- *$\beta$-robust if, for all demand matrices and for all assignment of predictions, the expected cost of the protocol is at most $\beta$ times the cost of an optimal optimum static SLN.*

## 3 OPTIMUM STATIC SKIP LIST NETWORK (OSSLN)

Given a set of node ids $V$ and a demand matrix $W$, the objective of the Optimum Static Skip List Network (OSSLN) problem is to find a height assignment $h : V \longrightarrow \mathbb{N}^+$ that minimizes the total communication cost (as defined in (4)) of the SLN $\mathcal{N} = (V, h, E)$ (as defined in (1)):

$$OSSLN(V, W) = \min_h \; Cost(\mathcal{N}, W). \tag{1}$$

In Schmid et al. (2016), the authors presented a dynamic programming algorithm to find the Optimum static distributed Binary Search Tree (OBST) in polynomial time ($O(n^3)$). This algorithm leverages a crucial property: once a vertex $v$ is chosen as the root of a tree induced by the vertices in a given interval, the subproblems induced by the vertex id intervals to the left and to the right of $v$ can be solved independently. The OSSLN problem exhibits a similar structure, but with a crucial difference.

If we define a **pivot vertex** $x \in [l, r]$, $1 \leq l \leq r \leq n$, such that $h_x = \max_{w \in [l,r]} h_w$, $x$ also partitions the OSSLN problem into two subproblems (on the intervals $[l, v)$ and $(v, r]$). However, a critical distinction arises when $h_x < \max_{w \in [l,r]} h_w$, in which case $x$ will not be a part of all the paths between vertex pairs $(u, v)$ $u \in [l, x), v \in (x, r]$. Specifically, for a vertex $x \in [l+1, r-1]$ such that $h_l, h_r > h_x$, any path between vertices $(u, v)$ where $u \leq l$ and $v \geq r$ does not include $x$. This changes the optimal substructure of the problem, requiring a more sophisticated dynamic programming approach.

Given a SLN $\mathcal{N}$, we define a **path** $P_h(l, r)$ to be the set of vertices in the *routing path* between the vertices $l$ and $r$ (as defined in (3)), excluding $l$, i.e., $l \notin P_h(l, r)$. This is a crucial simplification that

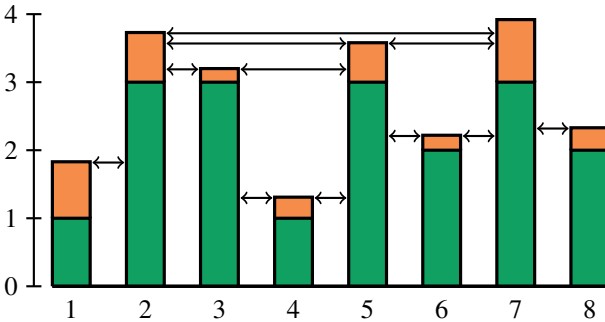

**Figure 2:** A Continuous SLN obtained by adding node-by-node the heights of the SLN in Figure **??** and the heights of the Continuous SLN in Figure 1. This is a possible outcome of LASLɪN where the predicted heights are in green and the uniform, random heights are in orange.

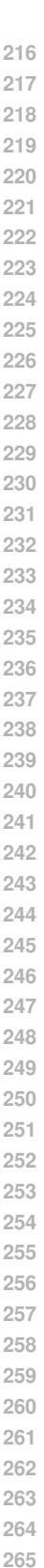
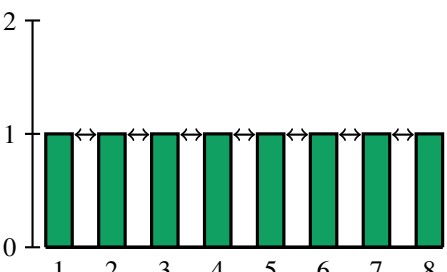
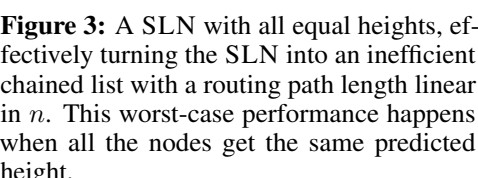
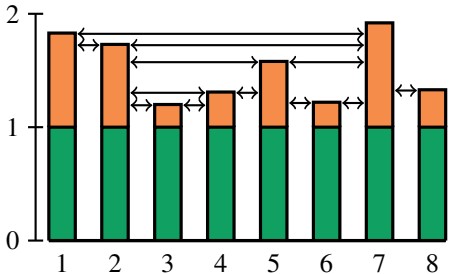

**Figure 3:** A SLN with all equal heights, effectively turning the SLN into an inefficient chained list with a routing path length linear in $n$. This worst-case performance happens when all the nodes get the same predicted height.

**Figure 4:** The addition of continuous, random heights restores the good performances.

allows the problem to be decomposed. Under this assumption, we have $d(l, r) = |P_h(l, r)|$. W.l.o.g., we assume that $P_h(l, r) = P_h(r, l)$.

Next, we define $L(x, h) = \min\{i \leq x \mid h_j \leq h_x, \ \forall j \in [i, x]\}$, i.e., the vertex of minimum id between the set of all (consecutive) vertices of height lower or equal to $h_x$, to the **left** of $x$, and, analogously, $R(x, h) = \max\{i \geq x \mid h_j \leq h_x, \ \forall j \in [x, i]\}$, i.e., the vertex with maximum id between the set of all vertices of height lower or equal to $h_x$, to the **right** of $x$. A vertex $x \in V$ belongs to a path $P_h(u, v)$ if and only if there does not exist any pair of vertices $(u', v')$ such that $u \leq u' < x, \ x < v' \leq v$, and (both) $h_{u'}, h_{v'} > h_x$.

Finally, given an arbitrary $x \in V$, we define $P_x(h)$ as the set of all pairs $(u, v) \in V \times V$, such that $x \in P_h(u, v)$.

**Lemma 1.** *If $h$ is a solution for the $OSSLN(V, W)$, then $\forall x \in V$ we have that:*

$$P_x(h) = \{(u, v) \mid (u < x) \wedge v \in [x, R(x, h)]\} \cup \{(u, v) \mid u \in [L(x, h), x] \wedge (v > R(x, h))\}. \quad (2)$$

Since $d(u, v) = |P_h(u, v)|$, we can now rewrite the total cost of an SLN $\mathcal{N} = (V, h, E)$ as:

$$\text{Cost}(\mathcal{N}, W) = \sum_{u, v \in V} W(u, v) \cdot d(u, v) = \sum_{u, v \in V} \sum_{x \in P_h(u, v)} W(u, v) = \sum_{x \in V} \sum_{(u, v) \in P_x(h)} W(u, v).$$

This means that the total cost is equal to the sum of contributions of all vertices $x \in V$, where the **contribution of a vertex** $x$ to the total cost is defined as:

$$\sum_{(u, v) \in P_x(h)} W(u, v) \quad (3)$$

**OSSLN with Bounded Height:** In the OSSLN with bounded height, the height assignment is of the form $h : V \longrightarrow [1, h_{\max}]$, where $h_{\max}$ is a parameter, typically bounded by $O(\log n)$ or $O(1)$.

Consider an interval $[l, r]$, $1 \leq l \leq r \leq n$ and a height bound $h \in [1, h_{\max}]$. Let $nl$, $1 \leq nl \leq l$, denote the leftmost vertex id, such that $\forall v \in [nl, l], h_v \leq h$, and, similarly, let $nr$, $r \leq nr \leq n$, denote the rightmost vertex id, such that $\forall v \in [r, nr], h_v \leq h$. Finally, let $\text{OSSLN}(l, r, nl, nr, h, W)$ denote the minimum total cost contributed by all vertices $v \in [l, r]$.

**Theorem 1.** *If $x \in [l, r]$ is a candidate pivot vertex with height $h \in [1, h_{\max}]$, then the following recurrence relation holds $\forall (nl, l, x, r, nr)$, $1 \leq nl \leq l \leq x \leq r \leq nr \leq n$:*

$$OSSLN(l, r, nl, nr, h, W) = \min \left\{ OSSLN(l, r, l, r, h - 1, W), \min_{x \in [l,r]} \left[ OSSLN(l, x - 1, nl, nr, h, W) + \right. \right.$$

$$\left. \left. OSSLN(x + 1, r, nl, nr, h, W) + \sum_{u < x} \sum_{v \in [x, nr]} W(u, v) + \sum_{u \in [nl, x)} \sum_{v > nr} W(u, v) \right] \right\}.$$

The complexity of the above dynamic programming algorithm is $\mathcal{O}(n^5 \cdot h_{\max})$, if we pre-compute an auxiliary matrix of aggregate weights $W'(x, y) = \sum_{z \leq y} W(x, z), \forall (x, y), 1 \leq x \leq y \leq n$.

**OSSLN with Unbounded height (OSSLNU):** We now present a dynamic programming algorithm for the OSSLN problem with unbounded heights, i.e., when the height assignment is of the form $h : V \longrightarrow [1, n]$. We show that, in this case, the algorithm's complexity can be reduced to $\mathcal{O}(n^3)$.

A crucial observation is that, for any solution to the OSSLNU problem, there exists a solution of cost less than or equal to that of the former, in which, $\forall [l, r], 1 \leq l \leq r \leq n$, there is a unique *pivot* vertex, i.e., a unique vertex whose height is the maximum height in that interval.

**Lemma 2.** *Let $h$ be an optimum solution to $OSSLNU(V, W)$. Then, there must exist a solution $h'$ such that $Cost((V, h', E'), W) \leq Cost((V, h, E), W)$, where, for any interval $[l, r], 1 \leq l \leq r \leq n$, there is exactly one vertex $x \in [l, r]$ with height $h'_x = \max_{w \in [l,r]} h'_w$.*

In OSSLNU, because we are only interested in the relative order of vertex heights, rather than in their absolute values, we have that $(nl = l)$ and $(nr = r)$, $\forall [l, r], 1 \leq l \leq r \leq n$, and the height parameter $h$ becomes unnecessary. This simplifies the recurrence relation, reducing the complexity to $\mathcal{O}(n^3)$.

**Theorem 2.** *Consider an interval $[l, r], 1 \leq l \leq r \leq n$. Let $OSSLNU(l, r, W)$ denote the minimum total cost contributed by all vertices $v \in [l, r]$. Let $x \in [l, r]$ be a candidate pivot vertex. Define $l$ as the leftmost index such that $\forall x, \in [l, r], \forall v \in [l, x], h_v \leq h_x$, and similarly define $r$ as the rightmost index such that $\forall v \in [x, r], h_v \leq h_x$. Then, the recurrence relation $OSSNU(l, r, W) =$*

$$\min_{x \in [l,r]} \left( OSSNU(l, x - 1, W) + OSSNU(x + 1, r, W) + \sum_{u < x} \sum_{v \in [x, r]} W(u, v) + \sum_{u \in [l, x)} \sum_{v > r} W(u, v) \right).$$

## 4 CONTINUOUS SLN

This section introduces a generalization of SLN called Continuous SLN.

**Definition 8** (Continuous SLN). *A Continuous SLN $\mathcal{N} = (V, h, E)$ is an SLN (Definition 1) except the height function $h$ is defined over the real numbers, i.e., $h : V \longrightarrow \mathbb{R}$.*

In this section, we show that Continuous SLN offer a set of properties similar to classic SLN with arguably simpler proofs. We provide the routines **join** and **leave** (Algorithms 2 and 4 in the Appendix) that are both core primitives of any P2P network, and we provide the routine **route** (Algorithm 1 in the Appendix) used by both data structure and P2P applications. We then present a simple Continuous SLN called UNIFORM whose heights are chosen uniformly at random in $(0, 1)$. Finally, we show that UNIFORM has performances matching those of skip lists at all levels as both a routing search path and the maximum degree are logarithmic with high probability.

**Protocol 1** (UNIFORM). *UNIFORM is a Continuous SLN that sets its heights to a real number in $(0, 1)$ chosen uniformly at random.*

**Theorem 3** (Expected Routing Path Length). *The routing path between any two nodes has length smaller than $2 \ln n$ on expectation.*

**Theorem 4** (Routing Path Length with High Probability). *The routing path between any two nodes has length smaller than $2e \cdot \ln n$ with probability greater than $1 - \frac{2}{n}$.*

**Theorem 5** (Expected Degree). *The expected degree of any node is lower than $4$.*

**Theorem 6** (Maximum Degree with High Probability). *The maximum degree is lower than $4 \ln n$ with probability greater than $1 - \frac{1}{n^3}$.*

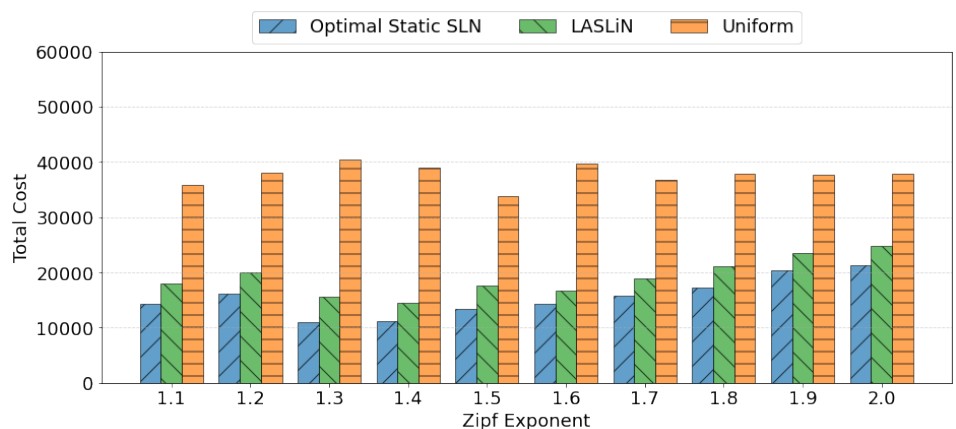

**Figure 5:** The costs of the Optimal static SLN, LASLıN with perfect predictions, and UNIFORM against various demand matrices with increasing Zipf parameter. LASLıN almost matches the optimal as it is $O(1)$-consistent. Very sparse demand matrices enable more efficient network optimization.

## 5 LEARNING-AUGMENTED CONTINUOUS SLN

In this section, we present LASLıN, a *learning-augmented* Continuous SLN protocol where each node can obtain a prediction of the future demand matrix at any time.

We first show that, for general demands, any demand-oblivious P2P protocol can be off by a factor of $O(\log n / \log \log n)$ compared to an optimum static SLN. This result provides a comparison point around which the consistency and robustness metrics of demand-aware algorithms can be evaluated.

**Theorem 7.** *We consider any P2P protocol that is not learning-augmented (Definition 6). Then, there exists a demand matrix $W$ such that the considered protocol has a cost of $\Omega(\log n / \log \log n) \cdot |W|$ in expectation while an optimal static* SLN *protocol pays $|W|$, where $|W|$ is the total sum of the elements of $W$.*

**Protocol 2** (LASLıN). LASLıN *is a Continuous* SLN *that takes an integer in $[1, \lceil \ln n \rceil]$ as a prediction.* LASLıN *initializes its heights by adding the input prediction with a real number in $(0, 1)$ chosen uniformly at random.*

**Theorem 8.** LASLıN *is a Continuous* SLN *that is $O(1)$-consistent and $O(\log^2 n)$-robust.*

We now define the notion of *super-sparse* matrix, which indicates a concentration of the weights of the matrix on a very small (logarithmic) number of coefficients.

**Definition 9** (super-sparse matrix). *A matrix $W \in M_n(\mathbb{R})$ is super-sparse if there exists a constant $c \geq 1$ and a restricted subset of node pairs $S \subseteq V^2$ such that:*

$$|S| \leq O(\log^c n) \qquad and \qquad \sum_{(u,v) \in S} W(u,v) \geq (1 - O(1/\log n)) \cdot |W|$$

**Theorem 9.** *Assume that the demand matrix $W$ is super-sparse. We consider the protocol Binary* LASLıN *(B-LASLıN) where the input prediction $p$ can only have two values, $1$ or $2$. B-LASLıN is $O(\log \log n)$-consistent and $O(\log n)$-robust.*

In asymptotics, the performances are always comparable to any demand-oblivious protocol (Theorem 7), with an exponential improvement in case of perfect predictions.

## 6 EXPERIMENTAL EVALUATION

**Demand matrices.** In our experiments, we generate communication demand based on the Zipfian distribution Adamic & Huberman (2002), with parameter $\zeta > 1$. Lower values of $\zeta$ result in a sparser demand matrix as the distribution becomes less skewed. The number of nodes in all experiments was set to $n = 32$, and each experiment was repeated 30 times.

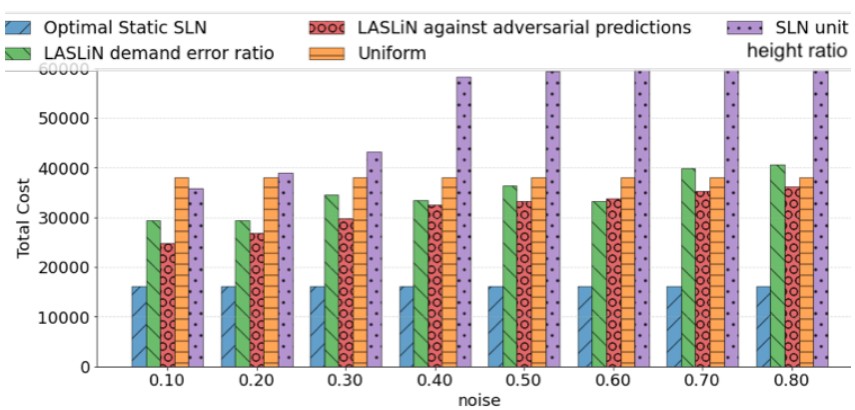

**Figure 6:** The costs of various protocols under different noises on the predictions.

**Noise generation.** We study the robustness of our algorithms by introducing noise in the predicted data Lykouris & Vassilvitskii (2018). The most used type of noise is additive, often gaussian Purohit et al. (2018) or lognormal Lykouris & Vassilvitskii (2018). Those types of noises however were unrelevant for our case as they almost did not impact the performance of our algorithms. Instead, we evaluate two noise models designed to simulate data imperfections. The first model, **Swap Noise**, simulates outdated communication data by replacing the communication value for each pair of vertices $(u, v)$ with a value from a different workload (with similar properties), an event that occurs with a probability equal to the noise percentage. This process mimics using stale information from a previous time step. The second model, **Prediction Noise**, introduces errors in vertex height prediction by resetting a vertex's height attribute to 1 with noise probability. This process mimics adversarial nodes in the network aiming to increase the communication path length.

The implementation of Swap Noise was adapted to the specific characteristics of each dataset. For the pFabric traces, where only a single instance of each trace was available, stale communication values were sourced from a different trace within the same collection; for instance, the communication value for a pair $(x, y)$ in the 0_1 trace might be replaced by the value for the same pair in the 0_5 trace. In contrast, for the Zipf workloads, which provided numerous instances for each distribution, a single workload was randomly selected at the outset of the experiment to serve as a fixed, global source of outdated communication values.

**Results.** In Figure 5 we evaluate the consistency of LASLıN in scenarios where the demand follows the Zipfian distribution, with varying parameter $\zeta \in [1.1, 2]$, and where the prediction advice is perfect (i.e., there is zero noise in the demand matrix). We compare the total communication cost (as defined in (4)) of LASLıN to that of the static optimum SLN and the demand-oblivious SLN (UNIFORM) baselines. We can see that, in contrast to the demand-oblivious SLN, LASLıN obtains close-to-optimum total communication cost in all workloads, regardless of their spatial locality, i.e., LASLıN is consistent.

In Figure 6 we evaluate the robustness of LASLıN in scenarios with increasing demand noise (x-axis). "LASLıN demand error ratio" corresponds to LASLıN under swap noise, "LASLıN against adversarial predictions" corresponds to LASLıN under prediction noise, finally, "SLN unit height ratio" corresponds LASLıN under prediction noise but without the random noise added. We can observe that the SLN performs increasingly worse (has higher cost) the greater the amount of change (noise) in the input demand matrix. LASLıN, on the other hand, achieves consistently lower communication cost (than the former), regardless of the type of noise and regardless of the prediction fault probability, i.e., LASLıN is robust relative to both type and magnitude of prediction errors.

## 7 RELATED WORK

**Improved designs of P2P networks.** P2P networks have been the focus of computer scientists for a long while (see Lua et al. (2005) for a detailed survey). In terms of P2P networks with constant degree,

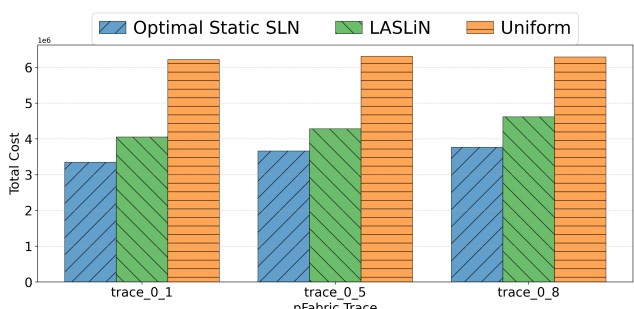

**Figure 7:** The costs of the optimal SLN, LASLıN and UNIFORM on three pFabric traces.

the notable contribution of Naor & Wieder (2007) that extends the work of Fraigniaud & Gauron (2006), giving a general P2P construction such that for a network with degree $d$ and $n$ nodes, you can have guaranteed path length of $\Theta(\log_d n)$. Lately, improving the design of networks given prior information about communication demand has grabbed the attention of researchers. In particular, the work of Avin & Schmid (2018) set the scene for demand-aware network design, which was then explored more with particular focus on datacenter networking de Oliviera Souza et al. (2022); Pourdamghani et al. (2023); Figiel et al. (2025).

Works closest to ours are by Avin et al. (2020c); Ciriani et al. (2007) in which the authors discuss a self-adjusting variant of skip list network: one that adjusts over time based on the incoming demand, which is in contrast with our work that focuses on an optimal skip list network with static height given the communication demand as the input. This is in particular important, as self-adjustments can introduce additional cost (e.g., in terms of delay) to the system. Also, the work of Mandal et al. (2015) discussed how a skip-list construction can be used for peer-to-peer communications, however they do not provide any theoretical upper bound for the communication time. Lastly, we point out to other extensions of skip list, namely Skip Graph Aspnes & Shah (2003) and SkipNet Harvey et al. (2003) that, to facilitate P2P communications, consider expected degree to be $O(\log n)$, in contrast to our design, which has a constant degree in expectation.

**Learning-augmented algorithms.** Given the rise of machine learning algorithms (that are prone to inherent unbounded failures), and after the pioneering work of Purohit et al. (2018) on ski rental & job scheduling, the focus has been shifted on studying consistency and robustness of algorithms augmented with the learned advice. Such an approach has been used in various applications so far[†], for example caching Lykouris & Vassilvitskii (2018), online TSP Gouleakis et al. (2023), list update Azar et al. (2025).

Some of the more recent works focus on improving single-source search data structures. Initially, the works of Lin et al. (2022); Chen et al. (2025) provided learning-augmented search trees. The works that are closest to our work are by Fu et al. (2025); Zeynali et al. (2024), which studied learning-augmented single-source skip lists. In contrast, we consider a more challenging model, where the focus is on pairwise communications rather than communication from a single source. Recently, we became aware of a not peer-reviewed manuscript Aradhya & Scheideler (2025) that aims to augment P2P networks with learned advice; however, this paper looks into improving self-stabilization and recovery of P2P networks, not improving communication delay.

## 8 CONCLUSION

In this work, introduced a continuous variant of Skip List Network (SLN), Continuous SLN. This variant is arguably easier to manipulate and analyze. In addition we show that we can use it to reach strong robustness properties even in an adversarial, fully distributed setting. Lastly, through experimental evaluation, we observed that our suggested algorithms outperform the theoretical results. As a future work, one can think of an alternative algorithm for OSSLN with a lower computational complexity.

---

[†]See a full list of recent works in the following website: `algorithms-with-predictions.github.io`

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

## 9 APPENDIX

Proof of Lemma 1.

*Proof.* If $u < x$ and $x \le v \le R(x, h)$, then, by definition of $R(x, h)$, $h_x \ge h_v \ge \min(h_u, h_v)$. Since $x$ lies between $u$ and $v$ and there are no edges "above" $h_x$ in the interval $[u, v]$ (i.e., $\nexists (w, z) \in E$, s.t. $h_w > h_x$ and $h_z > h_x$, $[w, z] \subseteq [u, v]$), $x$ must be a part of $P_h(u, v)$.

Similarly, if $L(x, h) \le u < x$ and $v > R(x, h)$, then, again by definition of $L(x, h)$, $h_x \ge h_u \ge \min(h_u, h_v)$. Since $x$ lies between $u$ and $v$ and there are no edges "above" $h_x$ in the interval $[u, v]$, $x$ must be a part of $P_h(u, v)$

It remains to show that no other pair $(u, v)$ belongs to $P_x(h)$. If $u \ge x$, then by definition of $P_h(u, v)$, $x \notin P(u, v)$. Otherwise, suppose $u < L(x, h)$ and $v > R(x, h)$. Let $u' = L(x, h) - 1$ and $v' = R(x, h) - 1$, by definitions of $L(x, h)$ and $R(x, h)$, we have that $h_x < h_{u'}$ and $h_x < h_{v'}$. Since $u \le u' < x < v' \le v$, we have that:

$$|P_h(u, u') \cup \{v'\} \cup P_h(v', v)| < |P_h(u, x) \cup \{x\} \cup P_h(x, v)|.$$

Because $P_h(u, v)$ is the shortest path from $u$ to $v$, it follows that $x \notin P_h(u, v)$. $\qquad \square$

Proof of Theorem 1.

*Proof.* The proof is by strong induction on the size of the interval $[l, r] \subseteq [1, n]$:

Note that, by definition, $nl = L(x, h)$ and $nr = R(x, h)$ for any possible $h$. Furthermore, by construction, for an interval induced by $l, r, nl, nr$ with bounded height $h$, assigning a pivot vertex with height equal to $h - 1$, we have $nl' = l$ and $nr' = r$.

**Base case** $(l = r)$**:** In this case, we have that $(x = l)$ and the set of paths $x$ belongs to is given by Lemma 1. By setting $nl = L(x, h)$ and $nr = R(x, h)$ in the proof, the vertex contribution to the total cost is given by:

$$\sum_{u < x} \sum_{v \in [x, nr]} W(u, v) + \sum_{u \in [nl, x)} \sum_{v > nr} W(u, v) \qquad (4)$$

**Induction step:** For $(h = 1)$, we can disregard the first half of the recursion, where we assign the pivot vertex a height lower than $h$. The total contribution of all vertices to the cost becomes:

$$\sum_{x \in [l, r]} \left( \sum_{u < x} \sum_{v \in [x, nr]} W(u, v) + \sum_{u \in [nl, x)} \sum_{v > nr} W(u, v) \right),$$

which, from the base case, is precisely what the second half of the recurrence equation will return for any choice of the pivot vertex.

Let $1 \le l < r \le n$ and $h \in [1, h_{\max}]$. Assume that the recurrence equation is true for all $h', l', r'$ such that $(h' < h) \wedge (r' - l') < (r - l)$. From previous step also assume that the equation is true for $(r' - l') = (r - l)$ with height $(h' < h)$.

Since, when assigning the pivot vertex with height lower than $h$, by construction, we have that $(nl = l)$ and $(nr = r)$. The minimum cost for the solution for any $nl$ and $nr$ is the minimum between keeping the height of the pivot, $(h_x = h)$, or decreasing it, $h_x \le (h - 1)$, which is given by the minimum between:

$$\min_{x \in [l, r]} \left[ \text{OSSLN}(l, x - 1, nl, nr, h, W) + \text{OSSLN}(x + 1, r, nl, nr, h, W) + \right.$$

$$\left. \sum_{u < x} \sum_{v \in [x, nr]} W(u, v) + \sum_{u \in [nl, x)} \sum_{v > nr} W(u, v) \right]$$

and $OSSLN(l, r, l, r, h - 1, W)$, which is precisely what we want to prove. $\qquad \square$

Proof of Lemma 2.

*Proof.* Suppose, for contradiction, that any possible optimal solution contains at least one interval $[l, r]$ with at least two vertices $x, y \in [l, r]$ with $h_x = h_y = \max(h_{l,r})$ and $x \neq y$.

Define $h'$ as follows for any vertex $v \in V$:

$$h'_v = \begin{cases} h_v + 1 & \text{if } v = x \text{ or } h_v > h_x \\ h_v & \text{otherwise} \end{cases}$$

Note that, while $h'_x > h'_y$, this height assignment is order preserving. If $h_a > h_b$ then $h'_a > h'_b$.

If $x \notin P_h(u, v)$, since we preserved the order between $x$ and all vertices taller than $x$, $x$ cannot appear as an obstacle between any such vertices. Analogously, since we preserved the order between all vertices with at most the same height as $x$ there can't be any new edges between vertices with at most the same height as $x$, so the path is unmodified.

If $x \in P_h(u, v)$, we know that:

$$P_h(u, v) = P_h(u, x) \cup \{x\} \cup P_h(x, v).$$

From our definition of path, we know that $x \notin P_h(x, v)$, and therefore $P_h(x, v) = P_{h'}(x, v)$. Now, let's consider a vertex $z$, such that $u \leq z < x$ and $h_z > h_x$. If such a vertex $z$ does not exist, then we can infer that $P_h = P_{h'}$. Furthermore, if there does exist such a vertex $z$, then:

$$|P_{h'}| = |P_h(u, z) \cup \{x\} \cup P_h(x, v)| \leq |P_h|.$$

Since $\text{Cost}((V, h, E), W) = \sum_{(u,v) \in V} W(u, v) \cdot d(u, v)$ and $d(u, v) = |P_h(u, v)|$, and any path in $h'$ is of at most the same size as in $h$, then, $\text{Cost}((V, h', E'), W) \leq \text{Cost}((V, h, E), W)$. Hence, the lemma holds. $\square$

Proof of Theorem 2.

*Proof.* The proof is by induction on the size of the interval $[l, r]$.

Note that, by Lemma 2, once a pivot vertex $x \in [l, r]$ is chosen as the highest vertex within the interval $[l, r]$, we can assume, without loss of generality, that all vertices in the optimal solutions to the left subinterval $[l, x - 1]$ and to the right subinterval $[x + 1, r]$ have heights strictly lower than $h_x$.

Therefore, for a solution $h$ to $\text{OSSLNU}(l, r, W)$, it follows that $l = L(x, h)$ and $r = R(x, h)$.

**Base Case** $(l = r)$: In this case, the interval contains a single vertex $x = l = r$. In this case, the cost reduces to the contribution of $x$ to the solution cost, which, from Lemma 1, and $L(x, h) = l$ and $R(x, h) = r$ as we demonstrated above, it follows that this expression corresponds exactly to:

$$\sum_{u<x} \sum_{v \in [x,r]} W(u, v) + \sum_{u \in [l,x)} \sum_{v>r} W(u, v) \tag{5}$$

**Inductive Step:** Suppose the recurrence holds for all subintervals of $[l, r]$ of size less than $(r - l + 1)$. For an interval $[l, r]$ of size greater than one, we consider a candidate pivot vertex $x \in [l, r]$.

The total cost of placing $x$ as pivot of the interval $[l, r]$ equals to the sum of:

- The optimal cost for the left subinterval $[l, x - 1]$,

- The optimal cost for the right subinterval $[x + 1, r]$,

- The cost of all paths that include $x$, which by Lemma 1 is given by the Eq.(5).

Since we minimize this total over all pivot choices $x \in [l, r]$, the recurrence holds. $\square$

Proof of Theorem 3.

*Proof.* Let $u$ and $v$ be two nodes. We note $X$ the random variable equal to the routing path length between $u$ and $v$. We note $X_{\text{up}}$ the number of height increases along the path, likewise $X_{\text{down}}$ is the number of path decreases. It holds that $X = X_{\text{up}} + X_{\text{down}}$. We will give an upper bound of $X_{\text{up}}$, the same upper bound will hold for $X_{\text{down}}$ by symmetry. Let $w$ be a node between $u$ and $v$ in the SLN, we note $1_w$ the indicator random variable equal to 1 if (1) $w$ is part of the path between $u$ and $v$, and (2) if the height increases when hoping to $w$. It holds $X_{\text{up}} = \sum_{w \in [u+1,v]} 1_w$.

We now compute $P[1_w = 1]$ for any $w \in [u+1, v]$. $w$ is part of the increasing path if and only if the height of $w$ is strictly greater than every other heights in $[u, w-1]$. The probability that two uniform random variables are equal is zero, hence there exists exactly one node with maximum height in $[u, w]$. The heights are independent and uniformly distributed, hence $w$ is the node with maximum height with probability $1/(|w - u| + 1)$:

$$\forall w \in [u+1, v], \quad P[1_w = 1] = \frac{1}{|w - u| + 1}$$

Noticing that $\mathbb{E}[1_w] = P[1_w = 1]$ and gathering the two previous equalities gives:

$$\mathbb{E}[X_{\text{up}}] = \sum_{w \in [u+1,v]} \frac{1}{|w - u| + 1} = \sum_{i=2}^{|v-u|+1} \frac{1}{i} \leq \ln n$$

The last inequality holds as the harmonic sum until $n$, $1 + 1/2 + \cdots + 1/n$, is lower than $1 + \ln n$. The claimed upper bound on the expected path length follows as the same reasoning can be done for $X_{\text{down}}$. □

Proof of Theorem 4.

*Proof.* Let $u$ and $v$ be two nodes such that $u < v$, and let $X_{\text{up}}$ be the random variable equal to the increasing routing path length between $u$ and $v$. Let $w \in [u, v]$, we note $1_w$ the indicator random variable equal to 1 if $w$ is part of the increasing path between $u$ and $v$ (see proof of Theorem 3).

We first show that all $1_w$ are independent random variables. Let $w$ and $w'$ be two nodes such that $u < w < w'$, it holds:

$$P[1_{w'} = 1 \mid 1_w = 1] = P[h_x < h_{w'} \forall x \in [w+1, w'-1] \cap \max_{y \in [u,w]} h_y < h'_w]$$

$$= P[h_y < h'_w \forall y \in [u, w'-1]]$$

$$= P[1_{w'} = 1]$$

Now that we showed independence, we use the following Chernoff's bound: For $Y$ the sum of independent Bernoulli random variables (a Poisson trial) and $\mu = \mathbb{E}[Y]$, $\forall \delta > 0$ it holds that:

$$P(Y \geq (1+\delta)\mu) < \left( \frac{e^\delta}{(1+\delta)^{1+\delta}} \right)^\mu$$

Replacing $Y = X_{\text{up}} = \sum_{w \in [u+1,v]} 1_w$, $\mu = \ln n$ and $\delta = e - 1$ gives:

$$P(X_{\text{up}} \geq e \cdot \ln n) < \frac{e^{\delta \cdot \ln n}}{(1+\delta)^{(1+\delta) \cdot \ln n}}$$

$$= \frac{n^\delta}{n^{(1+\delta) \cdot \ln(1+\delta)}} = \frac{1}{n}$$

By symmetry, the same result holds for $X_{\text{down}}$. Finally, we obtain the desired result with a union bound:

$$P(X \geq 2e \cdot \ln n) \leq P(X_{\text{up}} \geq e \cdot \ln n \cup X_{\text{down}} \geq e \cdot \ln n)$$

$$\leq P(X_{\text{up}} \geq e \cdot \ln n) + P(X_{\text{down}} \geq e \cdot \ln n)$$

$$< \frac{2}{n} \qquad \qquad □$$

Proof of Theorem 5.

*Proof.* If $u$ and $w$ are two nodes, we define $N_u$ the random variable equal to the number of neighbors of $u$, and $1_{u,w}$ the indicator random variable equal to one if $u$ and $w$ are neighbors. It holds $N_u = \sum_{w \neq u} 1_{u,w}$. Two nodes $u$ and $w$ with $u < w$ are neighbors if and only if $u$ and $w$ are the first and second highest nodes in $[u, w]$, hence it holds that $\mathbb{E}[1_{u,w}] = \frac{2}{|w-u| \cdot (|w-u|+1)}$. It therefore holds:

$$\mathbb{E}[N_u] = \sum_{w \in [1,n] \setminus \{u\}} \frac{2}{|w-u| \cdot (|w-u|+1)} \quad \leq 4 \cdot \sum_{i=1}^{n/2} \frac{1}{i \cdot (i+1)} = 4 \cdot \sum_{i=1}^{n/2} \frac{1}{i} - \frac{1}{i+1}$$

$$= 4 \cdot (1 - \frac{1}{n/2+1}) \leq 4 \qquad \square$$

Proof of Theorem 6.

*Proof.* Let $u$, $w$ and $w'$ be three different nodes. We name $1_{u,w}$ and $1_{u,w'}$ the indicator random variables equal to 1 if $u$ and $w$ are neighbors, and if $u$ and $w'$ are neighbors, respectively. We first show that $1_{u,w}$ and $1_{u,w'}$ are independent random variables: the proof is almost identical to the one of Theorem 4.

We first establish that each node taken individually has a degree lower than $\ln n$ with probability greater than $1 - \frac{1}{n^4}$. Let $N_u$ be the random variable equal to the number of neighbors of $u$, we use the Chernov bound for Poisson trials used in the proof of Theorem 4:

$$P[N_u \geq 4 \ln n] < \left( \frac{e^{(\ln n)-1}}{(\ln n)^{\ln n}} \right)^4 = \frac{(n/e)^4}{n^{4 \ln \ln n}} < \frac{1}{n^4}$$

This last equality holds when $n$ is greater than a given constant, which is not a problem as we are interested in cases when $n$ is large. We conclude with a union bound: the probability that at least one node has a degree greater than $4 \ln n$ is upper bounded by $\sum_{v \in V} P[N_v \geq 4 \ln n] \leq \frac{1}{n^3}$ and the claim holds. $\qquad \square$

Proof of Theorem 7.

*Proof.* As the considered protocol has a logarithmic maximum degree with high probability, there exists a constant $c$ such that the generated network has a maximum degree lower than $O(\ln^c n)$ with probability greater than $1 - O(1/n)$. Let $u$ be a node. We first cover the case when the maximum degree is lower than $\ln^c n$, all the considered probabilities and expected values are therefore conditional. In this case, we consider the disk centered around $u$ with smallest radius such that it contains no less than half of the nodes. The maximum distance between $u$ and a node in that disk is no lower than $O(\log n / \log \log n)$ as the network has a polylogarithmic maximum degree — this result can be easily obtained by upper bounding all the degrees by $O(\log^c n)$. There exists another node $v$ such that $v$ is at the border or outside of the disc with probability at least $1/2$. We consider the demand matrix that is zero everywhere except at the coefficient with row $u$ and column $v$. An optimal SLN protocol can clearly pay a cost of 1, the considered SLN therefore has an approximation ratio worse of $\omega(\log n / \log \log n)$ for that demand matrix.
Finally, we cover the cases where the maximum degree is higher than $\ln^c n$: That case only contributes with an additive constant to the distance between $u$ and $v$ as a distance is lower than $n$. We considered all the cases and the claim follows. $\qquad \square$

Proof of Theorem 9.

*Proof.* We start by proving that the maximum degree in LASLIN is lower than $4 \ln^2 n$ with high probability. Let $u$ be a node, let $p_u \in [1, \lceil \ln n \rceil]$ be the predicted height of $u$ and let $r_u \in (0, 1)$ be the uniform random noise added on top of $p_u$. We partition the neighbors of $u$ into $p_u$ subsets called $S(1), S(2) \dots S(p_u + 1)$: For all $i \in [1, p_u]$, $S(i)$ contains the neighbors with height between $i$ and $i + 1$, the last set $S(p_u + 1)$ contains the neighbors with height greater than $p_u + 1$. According to Theorem 6, each set from $S(1)$ to $S(p_u)$ contains fewer than $4 \ln n$ nodes with probability greater than $1 - \frac{1}{n^4}$ (this is the result before the last paragraph in the proof) — the set $S(p_u + 1)$ contains at most two nodes. We conclude with a union bound that $u$ has fewer than $4 \ln^2 n$ with probability

greater than $1 - \frac{1}{n^3}$, and that the maximum degree is lower than $4 \ln^2 n$ with probability greater than $1 - \frac{1}{n^2}$ also with a union bound.

The $O(1)$-consistency is direct as the comparative optimal static SLN has its heights within a $O(\log n)$ factor without loss of generality as its maximum degree is bounded by $O(\log n)$ by assumption. We now prove the robustness claim. A routing path between any two nodes $u$ and $v$ has two phases: first ascending (the heights always increase) and then descending (the heights always decrease). For both phase, the number of different encountered integer heights is upper-bounded by $O(\log n)$. For each encountered integer height, the expected routing path length is upper-bounded by $O(\log n)$ (Theorem 3). We finish the proof with an upper bound of each distance in the total weighted path lengths: $\sum_{u,v \in V} W(u,v) \cdot d(u,v) \leq O(\log^2 n) \cdot |W|$ and the optimal static SLN pays at least $|W|$. □

Proof of Theorem 1.

*Proof.* We start by proving that the maximum degree of each node is lower than $O(\log n)$ with high probability. As seen in the proof of Theorem 8, each node $u$ has fewer than $p_u \cdot \ln n$ neighbors with high probability, the claim follows since $p_u \leq 2$.

We prove that the expected total weighted path length is lower than $O(\log n) \cdot |W|$ regardless of the input predictions. The proof is similar to the proof of Theorem 8 except, as the predictions are upper-bounded by a constant, we avoid the additional $\log n$ factor.

Finally, we prove that the expected total weighted path length is lower than $O(\log \log n) \cdot |W|$. We consider the following set of predictions: let $S$ be the node pairs from Definition 9, we give prediction 2 to the nodes that have at least one coefficient from $S$ in their row or column in $W$, otherwise 1. We call $V_2$ ($V_1$) the set of nodes that have prediction 2 (1). As $W$ is super sparse, it holds that $|V_2| \leq O(\log^c n)$.

$$
\begin{aligned}
\sum_{u,v} W(u,v) \cdot d(u,v) &= \sum_{(u,v) \in S} W(u,v) \cdot d(u,v) + \sum_{(u,v) \in V^2 \setminus S} W(u,v) \cdot d(u,v) \\
&\leq \sum_{u,v \in V_2} W(u,v) \cdot d(u,v) + \sum_{(u,v) \in V^2 \setminus S} W(u,v) \cdot d(u,v) \\
&\leq O(\log \log n) \cdot |W| + O\left(\frac{1}{\log n}\right) \cdot |W| \cdot \ln n \\
&= O(\log \log n) \cdot |W|
\end{aligned}
$$

To go from line 3 to 4 we used (1) Theorem 3 on the routing paths between the nodes of $V_2$ — there are $O(polylog(n))$ of those nodes, they have the largest heights of the SLN, so a path has expected length $\ln O(polylog(n)) = O(\log \log n)$ — and on the routing paths from nodes in $V^2 \setminus S$, and (2) we used Definition 9 to upper-bound the sum of coefficients in $W$ outside of $S$. □

We present the internal variables for the three below routines Route, Join and Leave. A node $u$ holds the variables:

- $u$: the unique id of the node
- $h_u$: the height in $\mathbb{R}$ of node $u$
- $N_u$: the set of all neighbors of $u$
- $D_u$: dictionary mapping a neighbor $w \in N_u$ to its height
- $\text{anchor}_u$: the node with an id closest to $u$, either the successor or predecessor of $u$.

**Route.** The route algorithm (Algorithm 1) takes the destination node id $v$ and hops from node to node until it finds the destination node, returning $(v, True)$ or reaches a dead end in some node $w \neq v$, return $(w, False)$.

---

**Algorithm 1:** route($v$)

1   // code running at node $u$
2   **if** $u = v$ **then**
3      **return** $(u, \text{True})$
4   **else if** $u < v$ **then**
5      $S \leftarrow \{x \in N_u | u < x \leq v\}$
6      **if** $S = \emptyset$ **then**
7         **return** $(u, \text{False})$
8      **else**
9         $w \leftarrow \max S$
10        **return** $w.\text{route}(v)$
11   **else**
12      $S \leftarrow \{x \in N_u | v \leq x < u\}$
13      **if** $S = \emptyset$ **then**
14         **return** $(u, \text{False})$
15      **else**
16         $w \leftarrow \min S$
17        **return** $w.\text{route}(v)$

---

**Join.** The join algorithm (Algorithm 2) inserts the node inside the Continuous SLN based on its anchor local variable. The potential neighbor changes of the other nodes propagate through the network with the find_neighbors algorithm (Algorithm 3).

---

**Algorithm 2:** join()

1   // code running at node $u$
2   **if** *anchor* $< u$ **then**
3      $S \leftarrow \{x \in N_{\text{anchor}_u} | \text{anchor} < x\}$
4   **else**
5      $S \leftarrow \{x \in N_{\text{anchor}_u} | x < \text{anchor}\}$
6   **if** $S = \emptyset$ **then**
7      $N_u \leftarrow N_u \cup \text{anchor}_u.\text{find\_neighbors}(u, h_v, -\infty)$
8      Update $D_u$
9   **else**
10     **if** *anchor* $< u$ **then**
11        succ $\leftarrow \min S$
12     **else**
13        succ $\leftarrow \max S$
14     $N_u \leftarrow N_u \cup \text{anchor}_u.\text{find\_neighbors}(u, h_v, -\infty)$
15     $N_u \leftarrow N_u \cup \text{succ}.\text{find\_neighbors}(u, h_v, -\infty)$
16     Update $D_u$

---

**Leave.** The leave algorithm (Algorithm 4, in this paper's appendix) shows how an already-inserted node can leave the network. The potential neighbor changes propagate through the network with the delete_neighbors algorithm (Algorithm 5, in this paper's appendix).

---

**Algorithm 3:** find_neighbors($v$, $h_v$, current_height)

```
1  // code running at node u
```
2  **if** *current_height* $= -\infty$ **then**
3      $\text{anchor}_u \leftarrow v$
4  $N \leftarrow \emptyset$
5  **if** *current_height* $< h_u$ **then**
6      $N \leftarrow \{u\}$
7      $N_u \leftarrow N_u \cup \{v\}$
8      Update $D_u$
9  **if** $u < v$ **then**
10     $N \leftarrow \{x \in N_u | x \le v \text{ or } (h_v < D_u[x] \text{ and } h_v < h_u)\}$
11     **for** $x \in N_u$ **do**
12         **if** $x < u$ *and* $h_u \le D_u[x] \le h_v$ **then**
13             $N \leftarrow N \cup x.\text{find\_neighbors}(v, h_v, h_u)$
14 **else**
15     $N \leftarrow \{x \in N_u | v \le x \text{ or } (h_v < D_u[x] \text{ and } h_v < h_u)\}$
16     **for** $x \in N_u$ **do**
17         **if** $u < x$ *and* $h_u \le D_u[x] \le h_v$ **then**
18             $N \leftarrow N \cup x.\text{find\_neighbors}(v, h_v, h_u)$
19 **return** $N$

---

**Algorithm 4:** leave()

```
1  // code running at node u
```
2  **if** *anchor* $< u$ **then**
3      $S \leftarrow \{x \in N_{\text{anchor}_u} | \text{anchor} < x\}$
4  **else**
5      $S \leftarrow \{x \in N_{\text{anchor}_u} | x < \text{anchor}\}$
6  **if** $S = \emptyset$ **then**
7      $N_u \leftarrow N_u \cup \text{anchor}_u.\text{delete\_node}(u, h_v, -\infty)$
8      Update $D_u$
9  **else**
10     **if** *anchor* $< u$ **then**
11         $\text{succ} \leftarrow \min S$
12     **else**
13         $\text{succ} \leftarrow \max S$
14     $N_u \leftarrow N_u \cup \text{anchor}_u.\text{delete\_node}(u, h_v, -\infty)$
15     $N_u \leftarrow N_u \cup \text{succ}.\text{delete\_node}(u, h_v, -\infty)$
16     Update $D_u$

---

**Algorithm 5:** delete_neighbors($v$, $h_v$, current_height)

1 // code running at node $u$
2 $N \leftarrow \emptyset$
3 **if** *current_height* $< h_u$ **then**
4      $N \leftarrow \{u\}$
5      $N_u \leftarrow N_u \setminus \{v\}$
6      Update $D_u$
7 **if** $u < v$ **then**
8      $N \leftarrow \{x \in N_u | x \leq v \text{ or } (h_v < D_u[x] \text{ and } h_v < h_u)\}$
9      **for** $x \in N_u$ **do**
10          **if** $x < u$ *and* $h_u \leq D_u[x] \leq h_v$ **then**
11              $N \leftarrow N \cup x.\text{delete\_node}(v, h_v, h_u)$
12 **else**
13      $N \leftarrow \{x \in N_u | v \leq x \text{ or } (h_v < D_u[x] \text{ and } h_v < h_u)\}$
14      **for** $x \in N_u$ **do**
15          **if** $u < x$ *and* $h_u \leq D_u[x] \leq h_v$ **then**
16              $N \leftarrow N \cup x.\text{delete\_node}(v, h_v, h_u)$
17 **return** $N$

---

