# OpenReview forum: "A Learning-Augmented Overlay Network"
_ICLR.cc/2026/Conference — Submitted to ICLR 2026_

### Official Review · Reviewer_uwmG · 2025-10-25

**Soundness:** 2
**Presentation:** 2
**Contribution:** 2
**Rating:** 4
**Confidence:** 3

**Summary:**

This paper studies a routing problem in networks using so-called overlay networks. More specifically, they study Skip List Networks to route demands between nodes in a network. A SLN over n nodes is defined by assigning each node a height. These heights then imply edges and routing paths. The cost for routing a bit of demand between two nodes is equal to the length of the routing paths in the number of edges.
The goal is to find height such that the routing cost is not too big compared to the optimal heights for the given input.

There are three main contributions:
- A polynomial algorithm to compute the optimal heights and SLN for a given demand matrix via a dynamic program.
- A new P2P routing protocol via continuous skip lists: here the heights are sampled uniformly at random. This matches SOTA P2P protocols.
- Finally, the authors study this problem in a learning-augmented setting. They assume that they are given a prediction to achieve both consistency and robustness, that is, the performance if the prediction is correct or arbitrary. The prediction model is number per node, one can think of it as the optimal height for the coming demand. They present an algorithm that is $O(1)$-consistent and $O(\log^2 n)$-robust.

Moreover, the paper contains empirical experiments that show the benefits of the proposed algorithm under synthetic data (demand matrices and predictions).

**Strengths:**

- The paper is one of the first to study network routing in a learning-augmented setting.
- It gives a reasonable consistency-robustness tradeoff given the current SOTA bounds.
- Besides learning-augmented results, it also contains results for the offline and oblivious setting.
- The paper verifies its theoretical findings experimentally.

**Weaknesses:**

- A major weakness is that the paper only studies consistency and robustness. These two metrics on its own are not very meaningful, because predictions are rarely fully precise. It would have been more interesting to also analyze the performance w.r.t. a prediction error.
This massively limits the strength of the results.
- The first two results are rather offline/oblivious results and the connection to machine learning or learning representations are not clear. The contribution on learning-augmented algorithms also contains no lower bounds or further insights. I feel that the results are more preliminary and there are a lot of lose ends to be investigated in.

**Questions:**

L84 - L92: from the paragraph it becomes not clear what the prediction actually is. In such a paragraph I would expect to read "We consider a prediction that gives each node a number" or similarly.

Minor comments:
- L11: "which is _a_ natural extension"
- L23: performances are -> performance is
- L28: demonstrate -> demonstrates
- L64 "with with"
- L78 "based _on_ skip"
- L79 informations -> information
- L398: unrelevant -> irrelevant
- L479: "_we_ introduced"

---

### Official Review · Reviewer_5iBb · 2025-10-28

**Soundness:** 3
**Presentation:** 1
**Contribution:** 2
**Rating:** 2
**Confidence:** 2

**Summary:**

The paper considers routing in an overlay network.  A skip list network setup is used, where routing is determined by values assigned to each node, and the task is then to choose these values (and so the routing) so so as to minimise a weighted cost function (that weights flow rate by path length, akin to proportionally fair utility maximisation).

**Strengths:**

In my view the paper doesn't really have any real strengths as it stands.

**Weaknesses:**

The exposition in the paper is quite poor.  Not just in the justification of the work but especially on the experimental evaluation section where the optimimum static SLN is not defined nor is the rationale for the evaluation sufficiently explained or justified.   It justifies itself by reference to overlay networks, but I don't really buy that justification as relating to anything practical.   The paper is theoretical in nature, but its not clear to me that the theory contribution is itself timely and significant enough to warrant publication in ICLR.  As already noted, the evaluation leaves a lot to be desired.    I get the impression the paper was written in a hurry and I'd encourage the authors to take more time to re-write it properly.

**Questions:**

See weaknesses section above

---

### Official Review · Reviewer_K3gE · 2025-11-03

**Soundness:** 2
**Presentation:** 2
**Contribution:** 2
**Rating:** 2
**Confidence:** 2

**Summary:**

This paper focuses on the development and analysis of several P2P networks. In particular, recently it has been proposed to integrate ML techniques into skip list data structures at ICLR (Fu et al., 2025) and ICML (Zeynail et al., 2025). This paper extends those results to Skip List Networks (SLN), a type of P2P network. The paper presents a number of results on these networks, including formulating optimization problems, developing corresponding linear programming algorithms, etc.. Finally, they conduct an empirical analysis of their proposed techniques.

**Strengths:**

- The paper builds on very recent research developments on skip list data structures.
- This paper bridges theoretical results, such as those on skip list data structures and this paper, to novel networking applications.

**Weaknesses:**

- This paper does not seem appropriate for ICLR, it seems to be entirely a networking paper with the exception of a minor component of their last algorithm LASLIN is "learning-augmented" because it builds on the work of Fu et al. where ML techniques were integrated into list data structures (and further this aspect of LASLIN is not detailed).
- Related to the last point, the paper may be more easily understood by a networking audience, but I found it to not have necessary background information and missing important information to understand the paper.

**Questions:**

- Could you better explain how this fits into ICLR?

---

### Meta-Review · Area_Chair_dKnF · 2026-01-07

**Summary:**

All three reviewers questioned the paper’s suitability for ICLR. Three assigned low scores (2, 2, 4), criticizing it as primarily a networking paper with only superficial ML integration.

**Reviewer Concerns:**

The authors did not submit a rebuttal; therefore. None of the reviewers’ concerns were addressed during the discussion phase.

**Reviewer Scores:**

Reviewer K3gE (initial 2) did not engage further; score remains 2.
Reviewer 5iBb (initial 2) provided no follow-up; score remains 2.
Reviewer uwmG (initial 4) did not respond post-review; score remains 4.

---

### Decision · Program_Chairs · 2026-01-26

Reject